# Polysaccharide Based Implantable Drug Delivery: Development Strategies, Regulatory Requirements, and Future Perspectives

Sagar Salave [1], Dhwani Rana [1], Amit Sharma [1], K. Bharathi [1], Raghav Gupta [1], Shubhangi Khode [1], Derajram Benival [1,*] and Nagavendra Kommineni [2,*]

[1] Department of Pharmaceutics, National Institute of Pharmaceutical Education and Research (NIPER), Ahmedabad 382355, India
[2] Center for Biomedical Research, Population Council, New York, NY 10065, USA
* Correspondence: derajram@niperahm.res.in (D.B.); nagavendra.kommineni@gmail.com (N.K.)

**Abstract:** Implantable drug delivery systems advocate a wide array of potential benefits, including effective administration of drugs at lower concentrations and fewer side-effects whilst increasing patient compliance. Amongst several polymers used for fabricating implants, biopolymers such as polysaccharides are known for modulating drug delivery attributes as desired. The review describes the strategies employed for the development of polysaccharide-based implants. A comprehensive understanding of several polysaccharide polymers such as starch, cellulose, alginate, chitosan, pullulan, carrageenan, dextran, hyaluronic acid, agar, pectin, gellan gum is presented. Moreover, biomedical applications of these polysaccharide-based implantable devices along with the recent advancements carried out in the development of these systems have been mentioned. Implants for the oral cavity, nasal cavity, bone, ocular use, and antiviral therapy have been discussed in detail. The regulatory considerations with respect to implantable drug delivery has also been emphasized in the present work. This article aims to provide insights into the developmental strategies for polysaccharide-based implants.

**Keywords:** implants; polysaccharides; biopolymers; drug delivery; design; controlled release

## 1. Introduction

Implantable drug delivery systems offer wide therapeutic applications by providing targeted local delivery of drugs and long-term therapeutic effects. This effective delivery with lower drug concentrations results in minimizing the potential side-effects and ultimately enhancing the efficacy of treatment [1,2]. Drugs that would ordinarily be inappropriate for oral administration can be delivered as implants since, following implantation, the drugs would circumvent hepatic first pass metabolism and would also escape chemical degradation in the intestine and stomach, resulting in improved bioavailability. Controlled drug release over an extended period of time can be achieved by employing this system. The fluctuations in plasma drug concentrations such as attainment of peaks and valleys that occur from repeated intermediate dosing are avoided [2,3].

From the standpoint of patient compliance, the implantation might be relatively invasive, but good compliance can be achieved owing to a single-time implantation. Further, occurrence of any adverse effect that necessitates termination of treatment can be achieved by early removal of the implants. Moreover, hospitalization or persistent monitoring by healthcare staff might not be necessary for chronic conditions [2,3]. Figure 1 illustrates the advantages of an implantable drug delivery system. Marketed polysaccharides-based implants are enlisted in Table 1.

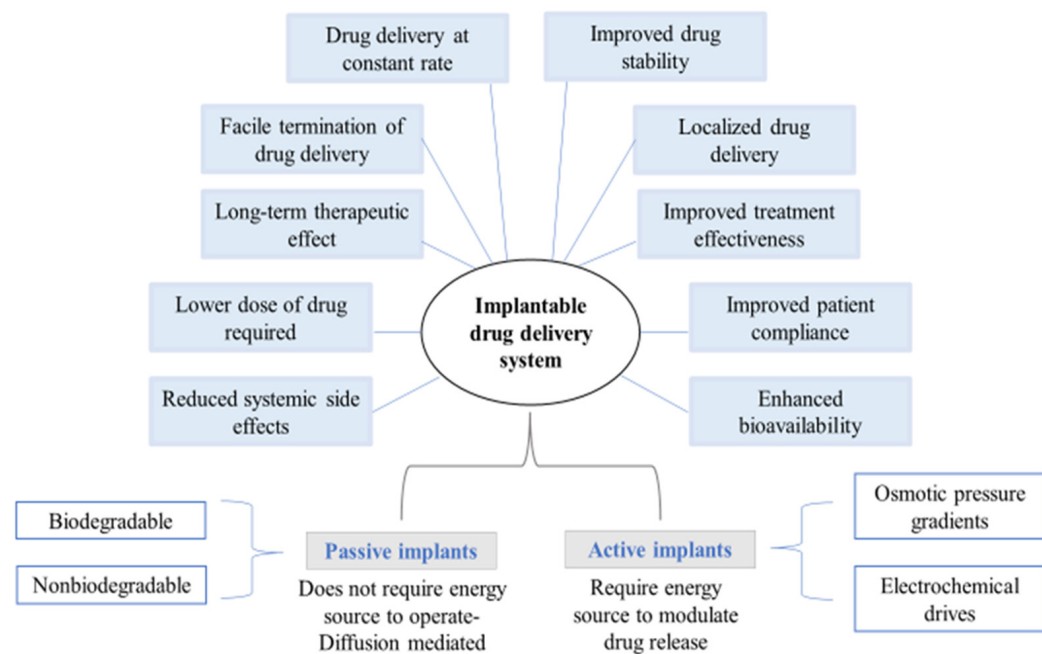

**Figure 1.** Design features of implantable drug delivery system.

**Table 1.** Polysaccharide based marketed implants [4,5].

| Name of the Marketed Ocular Inserts | Polysaccharides Used | Indication | Site of Implantation |
|---|---|---|---|
| LACRISERT® | Hydroxy propyl cellulose | Dry Eye disease (Keratoconjunctivitis Sicca) | Cul-de-Sac of the inferior eyelid |
| RETISERT™ | Microcrystalline cellulose | Chronic uveitis | Posterior region of the eye |

*Classification of Implantable Drug Delivery Devices*

Implantable devices for drug delivery can be broadly categorized into two major groups: active implants and passive implants. Passive implants depend upon a diffusion-controlled phenomena to achieve drug release, whereas active systems are dependent on energy that serves as the key driving factor to control drug release. Further, passive systems are biodegradable or non-biodegradable and are simple with no moving parts. They use different kinds of polymers that enable to achieve membrane-controlled drug release kinetics from the delivery systems. Active or dynamic drug delivery implants are relatively complex but offer greater control over the drug release [1,2].

Biopolymers arise from living organisms whose degradation products are not immunogenic. Biopolymers offer several benefits over synthetic polymers, including a well-organized structure, degradability, and renewability, all of which possess the ability to be utilised in the design of therapeutic devices like implants. A wide range of alternative uses of biopolymers are also evident, including their use as scaffolds for tissue engineering owing to its three-dimensional porous structures, as controlled or sustained release vehicles for drug delivery, and as temporary prostheses [6,7] (Figure 2).

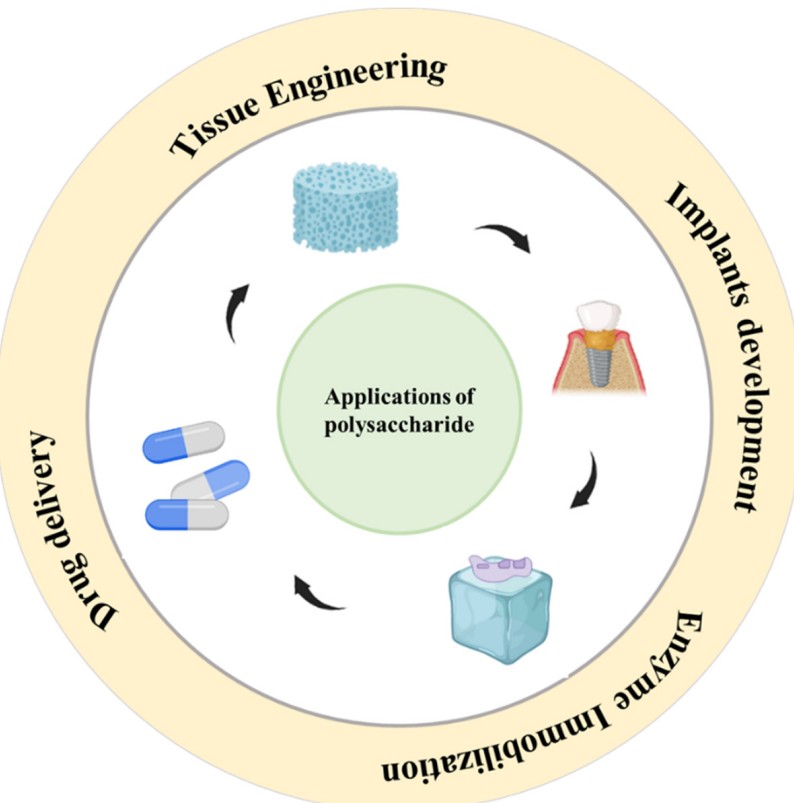

**Figure 2.** Drug delivery and biomedical applications of polysaccharides.

Polysaccharides belong to a diverse group of biopolymers comprising repetitive mono- or disaccharide units that are connected through enzyme-hydrolysable glycosidic linkages. These are widely used as controlled release drug carriers that impart remarkable physiological and physicochemical properties such as biocompatibility, biodegradability, and low immunogenicity [8]. Polysaccharide-based biopolymers are classified based on the source of their origin (Table 2).

**Table 2.** Classification of polysaccharide-biopolymers [7].

| Origin | Polysaccharides |
| --- | --- |
| Plant/algal | Starch (amylose/amylopectin), cellulose, agar, alginate, carrageenan, pectin, konjac, guar gum |
| Animal | Chitin/chitosan, hyaluronic acid |
| Bacterial | Xanthan, dextran, gellan, levan, curdlan, polygalactosamine |
| Fungal | Pullulan, elsinan, yeast glucans |

Moreover, advanced drug delivery systems based on polysaccharides can also improve the drug's pharmacokinetics due to their capacity to entrap the drug molecules in its interspaces, biocompatibility, and ability to provide a controlled release of the drug molecules [8]. All these characteristics make them ideal for use in implantable drug delivery. Figure 3 dictates the prime attributes of polysaccharides, making them ideal for the development of implants.

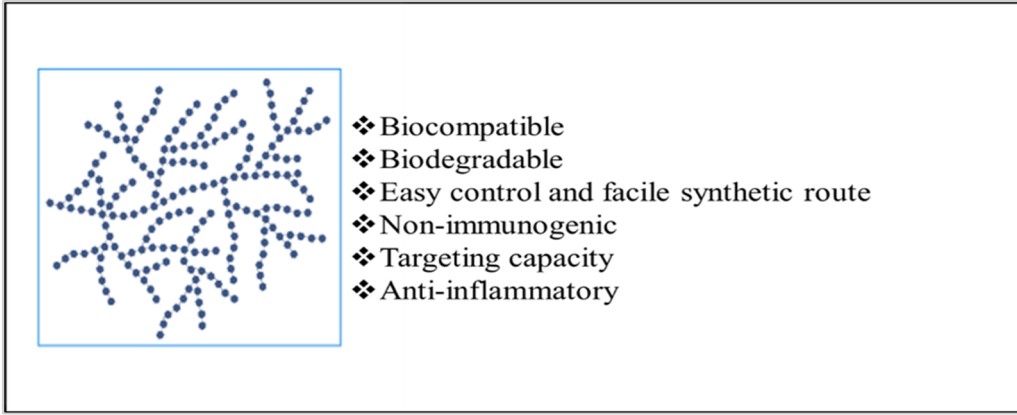

**Figure 3.** Unique characteristic of polysaccharides suitable for drug delivery and biomedical applications.

## 2. Strategies to Employ Polysaccharides in Implant Formulation

Implants can either be entirely made up of polysaccharides, contain polysaccharides as a part of the polymeric blend [9], or be coated with polysaccharides depending upon the therapeutic action required [10–12]. Figure 4 displays the representation of these strategies for developing polysaccharide-based implants. Kim et al. developed chitosan implants and evaluated them based on biodegradation and host tissue response in two different spinal cord rat models. Implants were placed in the intrathecal space of spinal cord for up to 6 months in the first model, whereas in the case of the second model, implants were directly placed into the spinal cord for up to 12 months. The results demonstrated the inertness of chitosan as a biomaterial due to the lack of elicitation of a chronic immune response making it suitable for long-term applications in the repair of spinal cord injury [13]. Nawrotek et al. fabricated and characterized hybrid implants composed of polycaprolactone/chitosan-hydroxyapatite for regeneration of peripheral nerves [14]. Norowski et al. made use of chitosan as a biodegradable coating material for implants to achieve a uniform delivery of antimicrobials at the local bone-implant interface, thereby avoiding initial bacterial attachment and subsequent growth during the early healing and osseointegration processes. The study concluded that chitosan coated implants have the potential to locally deliver antimicrobials that can inhibit bacteria without being toxic to host cells and tissues [15]. At times, modified polysaccharides have also been used for the coating of implants. For instance, Campelo et al. used sulfonate conjugated chitosan and dopamine-based coatings for metal implants that come in contact with blood for the purpose of modulating the potential surface interactions between blood and the biomaterial. Both sulfonated as well as native chitosan were grafted covalently onto the metallic devices by using PEG and dopamine as anchors. The study reported that sulfonate conjugated chitosan inhibited platelet activation, the process of blood coagulation, and also lowered calcification [16]. Employing these strategies ensures the design of implantable drug delivery systems with desirable and favourable outcomes.

Further, the type of release pattern required governs the choice of polysaccharide/combination of polysaccharides. For instance, Lacrisert is a marketed ocular insert to be placed in the cul-de-sac of the inferior eyelid and is indicated for the treatment of dry eye disease. The insert is entirely made up of hydroxypropyl cellulose (HPC), a cellulose-based polysaccharide, and dissolves gradually in the tear fluid [4]. The drug release from implants can also be modified by combining the matrix polymer with a secondary material, like a pore former, depending on the material properties of the implant. Certain polysaccharides are employed for their specific functions, e.g., water soluble cellulose-based polymers such as MC and HPMC are incorporated in the implants as pore formers. Aho et al. developed and studied the pore forming capacity of various grades of methyl cellulose and hydroxypropyl methylcellulose in extruded, drug-releasing polylactide (PLA) model implants. The pore forming polymers in the inert matrix, upon contact

with biological fluids, may leach out of the implant, thereby slowly increasing its porosity and simultaneously releasing the drug from the matrix [17].

**Figure 4.** Schematic representation of strategies employed to develop polysaccharide-based implants.

Moreover, the developmental strategy supremely depends upon the required action, and the choice of polymer is governed by their physiochemical properties. With the aim of improving the intraocular transparency of collagen matrices, Long et al. introduced HPMC into cross-linkable collagen to form HPMC-collagen composite membranes. The developed HPMC–collagen composite membranes proved to be effective for corneal regeneration [18]. Charlena et al. added HPMC to hydroxyapatite–chitosan composite as an injectable bone substitute. Bone filling material is injected in the form of a suspension to fill in the empty gaps of bone that occurred due to osteoporosis, and in this study, HPMC was used as a suspending agent [19].

## 3. Polysaccharide Based Polymers

### 3.1. Starch

Starch is an exceptionally versatile biopolymer that has been widely investigated in drug delivery owing to its low cost, biocompatibility, biodegradability, and tailorable characteristics. Two glycosidic polysaccharides, namely, amylopectin and amylose, form the structural composition of starch. The glucose units of amylose are bound in a linear fashion by $\alpha$-D-(1,4) glycosidic bonds, whereas amylopectin is a branched polymer, in which the branched chains of $\alpha$-D-(1,4) glucopyranose units are interconnected by $\alpha$-D-(1,6) glycosidic bonding. Based on its origin, the molecular mass of starch ranges around 50 to 500 million Da offering numerous applications in pharmaceutical formulation and technology [20]. Starch obtained from a variety of plant sources differs in physicochemical characteristics such as particle size distribution, crystallinity, and swelling power due to differences in the relative ratio of amylose to amylopectin content that can be attributed to environmental factors, genetic variations, and age of the plant [21].

Chemical modification of the surface-active hydroxyl functional groups in starch confers controlled release characteristics to it that aids in the efficient delivery of therapeutic molecules to the target site. The widely used methods of modification include esterification, etherification, oxidation, cross-linking, and cationization, that ultimately lead to alteration in the physicochemical properties of starch. Each glucose unit in starch consists of three free hydroxyl groups that, in the presence of oxidizing agents, under controlled temperature and pH conditions, yield carbonyl or carboxyl derivatives of reduced viscosity, high clarity, improved swellability, and better stability. Oxidation alters the surface morphology of the starch matrix, resulting in controlled biodegradation of the polymer and sustained release of the active agent over a specific period of time [22]. Etherification involves the alkali-catalyzed interaction of starch and alkyl oxides to yield hydroxyalkyl starches that promote disruption of internal bonds and enhance the freedom of motion in the amorphous region of starch granules. The weakening of bonds between starch chains depends on the degree of hydroxypropyl group substitution, which in turn enhances the

solubility, ease of hydration, swelling power, and enzyme digestibility. On subjection to freeze-thaw cycling, hydroxypropyl groups, being hydrophilic in nature, prevent the separation of water through syneresis, thus enhancing the freeze-thaw stability [23]. Acetylated starch is an example of ester modification, where the substitution by acyl groups confers hydrophobicity to the starch molecules and enhances their thermoplastic character. Hydrogen bond disorientation induced by acetylation in native starch, results in retardation of crystallization and a lowering of pasting temperature that in turn enhances its swelling power [24]. The introduction of ammonium, phosphonium, imino, or sulfonium groups confers a positive ionic charge on starche and this process is referred to as cationization. The modification method can be either dry or wet. Dry cationization involves spraying the cationic molecules on dried starch in the absence of a liquid medium, whereas wet cationization involves a liquid medium mediated reaction between starch and the cationic molecules. The characteristic properties exhibited by cationic starches include improved dispersibility, solubility, and stability [25]. Cross-linking of starch is facilitated by the use of polyfunctional reagents such as sodium trimetaphosphate, phosphorus oxychloride, genipin, and epichlorohydrin that form covalent bonds with the starch granules, making the ordering of the internal granules denser than its native counterparts. Cross-linking improves the tolerance of starch towards processes involving high shear and extreme pH [26]. Chemically modified starches are currently being investigated as a potential biomaterial for use in the design and development of drug delivery systems owing to their capacity to overcome the challenges associated with drug release, safety, cost, and storage.

### 3.2. Cellulose

Cellulose was first discovered by Payen in 1838, which was isolated from green plants and exhibits hydrophobic behavior. Cellulose is a polysaccharide that consists of a linear chain of about a hundred to over ten thousand D-glucose units held by $\beta$ (1→4) linkages. It is the most abundant biopolymer found exclusively in plant cell walls and is produced by certain species of bacteria, such as *Acetobacter xylinum*. Although the chemical structures of bacteria-based cellulose (BC) and plant-based cellulose (PC) are similar, BC does not require an extensive purification process because it can be produced without contamination of hemicelluloses and lignin. Owing to its mechanical strength and biocompatibility, it is widely used in tissue engineering. Even though both plant-based as well as bacteria-based cellulose are natural, considerable differences have been observed between them with respect to purity and macromolecular characteristics. In comparison to plant-based cellulose, bacterial cellulose has a high value of Young's modulus [27–29], a high-water absorption capacity, and a high aspect ratio in its fibers [30]. Cellulose degradability can be induced by oxidation, which is a very effective approach. Many different oxidizing agents, including $NaClO_2$, $CCl_4$, nitrogen oxides, and free nitroxyl radicals, can be used to create oxidized cellulose [31,32]. The remarkable properties such as high absorbability, antiviral and antibacterial effect, non-toxicity, and anti-adhesive qualities have made oxidized cellulose a popular choice for use as a wound healing material [33–35].

### 3.3. Alginate

Alginate was first discovered by E.C.C Stanford in 1881 [36]. Alginates are naturally occurring polysaccharides that are anionic in nature and are derived from the cell walls of brown algae such as *Laminaria hyperborea*, *Ascophyllum nodosum*, *Macrocystis pyrifera*, and several bacteria strains (Pseudomonas, Azotobacter). Alginates are linear biopolymers held by 1 → 4 glycosidic bonds comprising of 1,4 $\alpha$-L-guluronic acid (G) and $\beta$-D-mannuronic acid (M) residues arranged in the form of homogenous (poly-G, poly-M) or heterogeneous (MG) block-like patterns [37]. Extraction of alginate from seaweed is a multistage procedure. First, the dried raw material is treated with dilute mineral acid. Once the alginic acid becomes pure, it is converted into its water-soluble sodium salt in the presence of calcium carbonate, which is then transformed back into acid [38]. Alginate has great potential as a biomaterial for numerous biomedical applications, particularly in drug delivery,

wound healing, and tissue engineering. Characteristics like biocompatibility, mild gelling conditions, and simple modifications for preparing alginate derivatives with new properties make it suitable for these possible applications. Alginates undergo acid-mediated hydrolytic cleavage [38]. The reaction consists of three stages: (a) protonation of the oxygen atom at a glycosidic bond; (b) hydrolysis of the conjugate to generate the carboxonium ion and the non-reducing terminus; and (c) fast addition of water molecules onto the carboxonium ion, resulting in the formation of a reducing end. Sodium alginate can be stored as a dry powder at room temperature for several months without undergoing degradation [39].

### 3.4. Chitosan

Chitosan is a natural polysaccharide obtained by N-deacetylation of chitin. It is the second most abundant polysaccharide found in nature, after cellulose. The major source of chitin and chitosan include the shells of crustaceans such as crabs, prawns, and lobsters, but it is also obtained from the exoskeletons of various insects, mycelium of various fungi, including *Absidia glauca*, *Mucor rouxii*, *Gongronella butleri*, *Pleurotus sajor-caju*, *Aspergillus niger*, etc. Structurally, it is made up of randomly distributed N-acetyl-D-glucosamine (acetylated unit) and β-(1-4)-linked D-glucosamine (deacetylated unit) [40,41].

It is a polycationic, water-insoluble polysaccharide but is soluble in dilute organic acids like acetic acid, succinic acid, and formic acid. The physical properties of chitosan, like viscosity, are influenced by the degree of deacetylation, its molecular weight, temperature, and its pH [42]. It is a non-toxic, hydrophilic, biocompatible, and biodegradable polymer with good hydration capacity, which leads to its wide variety of applications in the pharmaceutical and cosmeceutical industries [42,43]. It is widely explored as a biomaterial for tissue engineering. It has also demonstrated its activity as a wound healing agent, antioxidant, and anti-obesity agent and also as an effective drug delivery carrier [44].

The low aqueous solubility and poor acid stability of chitosan limit its use in pharmaceutical products. Various structural modifications are performed at various functional groups for solubility enhancement of chitosan. Modifications at amino, hydroxyl or both amino and hydroxyl group forms N-modified, O-modified or N, O-modified chitosan derivatives. Incorporation of carboxymethyl groups at the C6-hydroxyl group or at the $NH_2$ functional group is an important method for enhancing chitosan's solubility [45,46].

### 3.5. Pullulan

Pullulan is a linear, unbranched, neutral exopolysaccharide obtained from the fungus *Aureobasidium pullulans* that is composed of maltotriose units interconnected by α-D-(1,6) glycosidic bonds. The three glucose units present in each maltotriose unit are in turn connected by α-D-(1,4) glycosidic bonds [47]. The distinctive linkage pattern of pullulan confers unique physicochemical properties to the biopolymer, including oxygen impermeability, adhesiveness, water solubility, and structural flexibility. The glycolipids present in the mycelia of fungus serve as zones for pullulan accumulation. Biosynthesis of pullulan involves uridine 5′-diphosphate glucose (UDPG) mediated formation of panosyl or isopanosyl residues linked to a lipid phosphate carrier that in turn undergoes polymerization to yield pullulan chains [48].

Based on the strain and fermentation parameters, the molecular weight of pullulan ranges from 45 to 600 KDa, offering polymers of varying viscosities, that play a crucial role in controlling the release of the active agent [49]. Mechanical stability of the pullulan matrix can be enhanced by inter-chain cross linking. In practice, the concentration of crosslinkers should be well optimized to obtain the desirable release as the water absorption capacity depends on the degree of cross-linking. Pullulan is soluble in water, partially soluble in dimethyl formamide and dimethyl sulfoxide, and insoluble in other organic solvents. Imparting hydrophobicity to pullulan by chemical modification is highly desirable for its use in drug delivery applications. Grafting is one such method that involves either the free-radical mediated covalent attachment of hydrophobic monomers onto a polymeric backbone or the incorporation of pre-synthesized grafts to the polymeric chain charged

with complementary functional groups [50]. Grafting of methyl acrylate onto pullulan by means of copolymerization, with ceric ammonium nitrate as the initiator, was found to decrease the hydrophilicity. Owing to an increase in the total number of hydrophobic monomers with an increase in the percentage grafting, a decline in the water absorption capacity was observed by Shengjun et al. [51].

The introduction of negative charge in the modified derivative following carboxymethylation prolongs the retention time of the polymeric matrix inside the body. The carboxymethylation of pullulan through an alkali-mediated reaction with sodium chloroacetate in a 2:1 isopropanol water mixture revealed that the substitution of the hydroxyl group is predominant at C-2. The order of reactivity was found to be OH-2 > OH-4 > OH-6 > OH-3 [52]. The above-mentioned approaches for developing multifunctional pullulan biopolymers enable its wide applicability in tissue engineering, targeted theragnostic, and gene delivery.

### 3.6. Carrageenan

Carrageenan is a marine polysaccharide isolated from certain species of red seaweed belonging to the class Rhodophyceae. It is a linear, anionic, sulfated polysaccharide composed of alternative units of D-galactose and 3,6-anhydro-galactose interconnected by alternate α-(1,3) and β-(1,4) glycosidic bonds. The average molecular weight is above 110 KDa, in which the ester-sulfate content contributes to around 15 to 40% of the total content [53]. Gelation of carrageenan occurs on cooling in the presence of cations like Na+ and K+. Based on the number of ester-sulfate groups, its relative position, and the subsequent differences in solubility, carrageenan can be categorized into three main classes, namely, kappa (κ), iota (ι), and lambda (λ). The respective ester-sulfate content of κ, ι and λ was found to be 20%, 33%, and 40%, which is prone to variations attributable to the differences in the species of the seaweed. Carrageenan with a higher ester content is characterized by low solubility temperature and gel strength [54]. In order to achieve control over the gelation properties and to overcome the challenges associated with the degradation of the polysaccharide on exposure to physiological conditions, various physical and chemical modification approaches have been investigated. Cross-linking between the charged polymer and counterions through the formation of physical bonds results in brittleness of the matrix. Hence, chemical cross-linking is mostly employed in drug delivery and tissue engineering applications, owing to its ability to form stable covalent bonds. Methacrylation of carrageenan has received wide attention as it confers the ability to be photocrosslinked. Silvia et al. in the presence of ultraviolet light and a chemical photo initiator, developed photocrosslinkable hydrogels of κ-carrageenan by varying the degree of methacrylation that enabled easy tailoring of properties such as viscosity, swelling ratio, elastic moduli, and pore size distribution [55]. However, in order to overcome the drawbacks associated with UV-crosslinking, photoinitiators that can be activated by visible light, such as Eosin Y and triethanolamine, have been investigated. Other modification approaches include de-esterification, carboxymethylation, thiolation, acetylation, and phosphorylation. Besides its biocompatibility and biodegradability, carrageenan possesses unique immunomodulatory properties that enable it to exert protective action against viral infections. The anti-viral activity is related to the degree of sulphation, location of the sulfate groups and molecular weight, and it has been identified that λ-carrageenan exhibited a greater inhibitory effect against drug-resistant viruses. Carrageenan also exhibits antimicrobial effects against foodborne pathogenic bacteria such as Staphylococcus aureus. The sulfate groups in carrageenan facilitate its binding to biologically active proteins that are responsible for its anticoagulant activity [56]. Prolonged drug release from the carrageenan matrix can be achieved as a result of gradient hydration on exposure to an aqueous environment that causes the matrix to swell, forming a gel or viscous outer layer. This layer acts as a polymeric shell to control the dissolution and diffusion of the loaded drug. The release rate is affected by the type of carrageenan used, and it has been found that κ-carrageenan exhibits a faster release rate. Its adhesiveness to mucosal and epithelial tissues serves as an

added advantage for prolonging the drug release. Hence, carrageenan can be widely used in drug delivery applications due to its above-mentioned versatile characteristics [57].

### 3.7. Dextran

The term Dextran was coined by Scheibler in the year 1874, during his investigation of gelatinous thickening (so-called frog's spawn) produced during the production of beet-sugar juices [58]. Dextran is an extracellular water-soluble polysaccharide (EPS) that is produced by the lactic acid bacteria (LAB), *Leuconostoc mesenteroides*, in a sucrose rich media [59,60]. Dextran is a high molecular weight polysaccharide having a molecular weight of up to 440 MDa. Structurally, it is composed of D-glucose units linked by α-(1-6) bonds and with branches linked by α-(1-4), α-(1-3) and α-(1-2) bonds [60].

Dextran possesses various important properties that make it a useful biopolymer for the development of implantable systems. It is a non-immunogenic, biocompatible, and biodegradable polymer with excellent solubility characteristics. It is generally not degraded when acted upon by the enzymes present in the upper gastrointestinal tract. The activity of dextranase enzymes located in the lumen of the large intestine, liver, kidney, and spleen is responsible for its degradation [61,62]. The molecular weight and branching of dextran dictate its rheological properties. Dextran, having a molecular weight in the range of 40,000 to 2 million, behaves like a Newtonian system up to a concentration of 30% [63]. High molecular weight Dextran shows pseudoplastic behavior. This can be explained by the fact that the forces generated during the application of shear would have broken the structural interaction between α-glucan chains in solution [64].

Dextran has been subjected to various structural modifications to enhance its physicochemical properties. The presence of numerous hydroxyl (-OH) groups and narrow molecular weight distribution makes it a potential biopolymer for chemical modification. Various derivatives of dextran are synthesized by etherification, esterification, and oxidation reactions. An important derivative of dextran is dialdehyde dextran that is prepared by the periodate oxidation method [65]. It has been conjugated with protein, and this dialdehyde dextran–protein conjugate is widely studied as a carrier molecule in immunodetection techniques [62].

### 3.8. Hyaluronic Acid (HA)

In 1934, scientist Karl Meyer and his assistant John Palmer isolated a new high molecular weight (~4 million Da) polysaccharide from vitreous humor of bovine animals. They found that this polysaccharide contained an amino sugar and uronic acid. The scientist named this new polysaccharide hyaluronic acid, derived from the words "hyaloid" (vitreous) and "uronic acid". It took nearly 25 years of the research work to elucidate the structure of repeating disaccharide unit that formed this polysaccharide [66].

HA consists of D-glucuronic acid and D-N-acetylglucosamine which are linked to each other through alternating β-1,4 and β-1,3 glycosidic bonds. This polysaccharide has been known to perform various significant roles in microorganisms and higher animals, including human beings [67]. It is synthesized naturally by a class of integral membrane enzymes that are present on the inner surface of the plasma membrane, called HA synthases. It is degraded in the body via two major ways: one is by a class of enzymes called hyaluronidases which act by hydrolysing the β-(1,4) linkages between N-acetyl-D-glucosamine and D-glucuronic acid residues; the other is by a free radical mechanism in the presence of several reducing agents like ascorbic acid, cuprous ions, etc. [68]. In humans, it is present in its salt form, i.e., hyaluronate, and is found in a very high concentration in skin, umbilical cord, and vitreous humor. Because of its various important properties, including excellent viscoelasticity, biocompatibility, mucoadhesiveness, and high moisture retention ability, it has found a wide variety of applications in cosmetics, nutraceuticals, and pharmaceuticals [69].

HA is a weak polyelectrolyte, and therefore, its rheological properties in aqueous solution are affected by varying conditions of pH, ionic strength, and temperature. The change in pH of the aqueous buffered HA solution significantly affects its stability. HA degradation occurs at acidic (pH < 4) and alkaline (pH > 11) conditions [70]. At very high concentration, the HA solution has a very high viscosity, but on application of pressure, it flows easily. This shear-thinning behavior is due to the breakdown of intermolecular H-bonds and hydrophobic interactions under increasing shear rates [67,71,72].

Various chemical modifications have performed on HA to develop HA derivatives with desired characteristics. HA conjugates have also been developed by various researchers. This crosslinking of HA with a wide variety of molecules leads to the development of carrier systems with improved drug delivery properties [72].

### 3.9. Agar

Agar was first found in Japan in 1958, called agar-agar. It was first introduced as phycocolloid. Innkeeper Minoya Tarozaemon accidentally discovered the manufacturing process of agar [73]. Agar is a gel-forming agent found in red seaweed. It is a mixture of two polysaccharides; a naturally gelling fraction called agarose, around 70% of total, and agaropectin, a sulphated nongelling fraction. Agarose is a neutral gel-forming molecule. Due to a similarity in their backbone structures, agaropectin and agarose are closely linked. The bacterial species do not enzymatically breakdown agar. Agarose and agaropectin consist of chemical structure alternating between 1,3-linked-B-D-galactopyranose, and 1,4-linked-3,6-anhydro-L-galactopyranose which can be masked to variable degrees by various sugar residues. Agarose is the component with the highest gelling tendency. Agar gels can withstand temperatures of up to 65 °C, but molten agar cannot gel until cooled to roughly 40 °C. Gels made of agar are very transparent [74]. Agar's exceptional ability to gel is solely due to hydrogen bonds created between its linear galactan chains, which offer high reversibility in gelling and melting points that differ by roughly about 45 °C. Agar was broken down by enzymes and acid to produce agarobiose and neoagarobiose, respectively, which shows that 1, 3-linked-β-D-galactopyranose and 1, 4-linked-3, 6-anhydro-α-L-galactopyranose alternate with agarobiose repeating disaccharide units to make up agarose. Agaropectin has a substantial number of acid groups, such as sulphate, pyruvate, and glucuronate groups, despite having what seems to be the same structural backbone as agarose.

Agar can be found as a coarse or fine powder, or as translucent, flavorless, odorless strips. It could be a weak yellowish-orange, a greyish yellow with a pale-yellow tint, or it could be colorless. Agar is brittle when dry and rough when damp. Solubility is almost insoluble in cold water and ethanol (95%), but soluble in hot water to form a viscous solution. On chilling, 1% $w/v$ aqueous solution of agar transforms into a rigid jelly. Agar is generally recognized as safe-GRAS certified, accepted in Europe for use as a food additive. Agar is also a component of the United States Food and Drug Administration (USFDA) Inactive Ingredients Database (oral tablets) and included in the acceptable non-medicinal ingredients in Canada. It is also included in non-parenteral medications with UK licensing [20]. Agar gels are thermo-reversible when cooled below the temperature at which homogenous hot water solutions gel. Agarose, the gelling component in agars, which is basically a neutral polymer, does not need specialized counterions to produce gels, in contrast to anionic more sulfated agars. These gels are fragrance free and tasteless, have a significant temperature hysteresis between the gelling point (Tg) and melting points (Tm), are cloudy, fragile, and syneresis-prone.

### 3.10. Pectin

Pectin is a carbohydrate-like plant element with a high molecular mass that is composed mostly of chains of galacturonic acid units connected as 1,4-a-glucosides. As per USP 32 pectin is defined as a purified carbohydrate product derived from a dilute acid extract of the inner portion of the rind of citrus fruits or apple pomace [20].

Pectic acids are poly (α-D-galactopyranosyluronic acids) galacturonoglycans that lack or have a very low concentration of methyl ester groups. There are several levels of neutralization for pectic acids. Pectates are the name for the salts of pectic acids [75]. Pectin is a high-value functional food component that is frequently used as a stabilizer and gelling agent. Additionally, it is a plentiful, common, and multipurpose component of all terrestrial plant's cell walls. Pectin defines a family of oligosaccharides and polysaccharides that share similar properties, yet are exceedingly different in their fine structures [76].

Homogalacturonan (HG), which makes up around 60% of the total quantity of pectin, is the most prevalent kind of pectin in cell walls. The HG polymer has an α-1,4-linked galacturonic acid residue backbone. For citrus, sugar beet, and apple pectin, this backbone's minimum predicted length is between 72 and 100 galacturonic acid residues. This backbone's galacturonic acid moieties can be O-acetylated at O-2 and/or O-3 and/or methyl esterified at C-6. For instance, pectin molecules that are sensitive to $Ca^{2+}$ cross-linking are produced by blocks containing more than 10 non-esterified galacturonic acid residues [77].

### 3.11. Gellan Gum

Gellam gum (GG) is a linear, negatively charged exopolysaccharide, also called as S-60 [78]. A bacterial polysaccharide called GG was initially made commercially by Kelco (now Monsanto PLC) using the bacterium Sphingomonas elodea. The de-esterification produces a stiff, brittle gel, while the native polysaccharide generates a weak, elastic gel [79]. The backbone of natural gellan is composed of a repeating unit of β-1,3-D-glucose, β-1,4-D-glucuronic acid, β-1,3-D-glucose, and α-1,4-L-rhamnose. Two acyl groups, acetate and glycerate, are joined to the glucose residue adjacent to glucuronic acid. The proportions of the three major components are roughly as follows: 60% glucose, 20% rhamnose, and 20% glucuronic acid [80]. Commercially, GG is offered in two different forms, the first of which is sold under the trade name Gelrite[TM] and is also referred to as acylated GG or high acyl GG, and the second is sold under the trade name Kelcogel[TM] and is referred to as low acyl GG or deacylated GG [78]. The natural GG is water soluble; it has a high viscosity at low concentrations and a rheological yield point. The viscosity is steady in the pH range of 3–11 and the temperature range of 20–700 °C. A reversible drop in viscosity is seen on a 0.5% solution in water at about 700 °C, coinciding with the conformational change. This results in the formation of an elastic and soft thermo-reversible gel with the same melting and setting temperatures and no hysteresis from a 1% solution in water heated to 90 °C for 10 min [81]. At high temperatures, gellan molecules appear as random coils, but at low temperatures, they appear as double helices. GG can withstand acid and heat stress when being manufactured. It is ductile, non-toxic, biocompatible, biodegradable, and thermoresponsive. GG possesses mucoadhesive qualities. Due to its negative charge, this polysaccharide can create polyelectrolytes with polymers that have the opposite charge, such as chitosan. GG is regarded as a pseudoplastic at high shear rates. GG is not destroyed by an acidic environment and is resistant to enzymatic activity. The GG beads expand at high pH levels and remain stable at low pH levels. Additionally, it possesses a broad range of mechanical, acceptable rheological, and high processability qualities. The finest qualities of GG are its high efficiency, malleability, and gelling ability [78].

Advantages and disadvantages of various polysaccharides are enlisted in Table 3 whereas polysaccharide derivatives and their modified properties are summarized in Table 4.

**Table 3.** Advantages and disadvantages of polysaccharides.

| Polysaccharide | Advantages | Disadvantages | References |
|---|---|---|---|
| Starch | <ul><li>Biodegradable</li><li>Wide-abundance in nature</li><li>Renewable</li><li>Low material cost</li><li>Easy processability</li><li>High viscosity and molecular weight confer controlled release characteristics</li><li>Enhanced freeze–thaw stability on etherification</li></ul> | <ul><li>Poor mechanical properties</li><li>Brittleness</li><li>Non-compatible with hydrophobic polymers</li></ul> | [82] |
| Cellulose | <ul><li>Most abundant polysaccharide</li><li>High water holding capacity</li><li>Biocompatibility</li><li>Good swelling index</li><li>Diffusion-controlled release from the matrix</li><li>Occurs as different morphological forms such as microfibril/nanofibril, micro/nanocrystalline that finds numerous applications in tissue engineering</li><li>Inherent anti-bacterial and wound healing properties</li></ul> | <ul><li>Lack of thermoplasticity</li><li>Poor crease resistance</li><li>Prone to antigenicity</li></ul> | [83] |
| Alginate | <ul><li>Non-antigenicity</li><li>Biocompatible</li><li>Chelating ability with divalent cations</li><li>Structural and chemical resemblance to extracellular matrix proteins</li><li>Long term storage stability</li><li>Tunable mechanical properties</li><li>Attenuates chronic inflammation and decreases oxidative stress</li></ul> | <ul><li>Susceptible to hydrolysis in acidic environment</li><li>Fabrication is challenging due to its low solubility</li></ul> | [84] |
| Chitosan | <ul><li>Biocompatible</li><li>Good hydration capacity</li><li>Mucoadhesive property</li></ul> | <ul><li>Poor solubility at physiological pH</li></ul> | [42,43] |
| Pullulan | <ul><li>Impermeable to oxygen</li><li>Non-hygroscopic</li><li>Thermal stability</li><li>Bioadhesiveness</li><li>High structural flexibility</li><li>Film forming ability</li><li>Non-immunogenic</li><li>Anti-static and elastic properties enable processing as compression moldings</li></ul> | <ul><li>Highly expensive</li><li>Brittle</li><li>Low mechanical strength</li></ul> | [85] |
| Carrageenan | <ul><li>Immunomodulatory properties that protect against viral infections</li><li>Bioadhesiveness</li><li>pH/temperature sensitive delivery</li></ul> | <ul><li>Low gel strength</li><li>Extra care is required when processing into blood-contact biomaterials such as tissue engineering scaffolds due to its anti-coagulant activity</li></ul> | [57] |

**Table 3.** *Cont.*

| Polysaccharide | Advantages | Disadvantages | References |
|---|---|---|---|
| Dextran | • Non-immunogenic<br>• Biocompatible<br>• Excellent solubility characteristics | • High cost<br>• Non-availability | [62] |
| Hyaluronic acid | • Excellent viscoelasticity<br>• Biocompatibility<br>• Mucoadhesive<br>• High moisture retention ability | • High cost<br>• Poor mechanical property | [69] |
| Agar | • Thermo-reversible gelation behavior<br>• Excellent biocompatibility | • Fragile<br>• Syneresis-prone | [86] |
| Pectin | • Gel-forming property<br>• Mucoadhesive property<br>• Anti-metastatic activity<br>• Excellent emulsifying and film-forming property | • Poor water vapor barrier<br>• Low mechanical property | [87] |
| Gellan gum | • Non-toxic<br>• Thermo-responsive<br>• Mucoadhesive property<br>• Significant gelling ability | • Relatively expensive | [78] |

**Table 4.** Polysaccharides derivatives and their modified properties.

| Polysaccharide | Derivatives | Modification in Properties | References |
|---|---|---|---|
| Chitosan | Carboxymethyl chitosan | Enhanced solubility, water retention capacity and antioxidant activity | [45] |
| | N-Trimethyl Chitosan | Increased mucoadhesive property | |
| | Thiolated Chitosan | High permeation, mucoadhesion, higher solubility at physiological pH, in situ gelling property | |
| | Polyethylene glycol (PEG)-grafted chitosan | Increased solubility over a wide range of pH, enhanced mucoadhesion | |
| Hyaluronic acid (HA) | HA esters | Decreased water solubility of HA, with the aim to reduce its susceptibility to hyaluronidase degradation and enhance its in-situ residence time | [72,88] |
| | HA amides | High grafting yield | |
| | Can be conjugated with various biocompatible polymers like PEG, Chitosan, Poly(lactide-co-glycolide) | Delivery carrier for various hydrophobic and hydrophilic drugs | |
| Dextran | Dextran esters | Greater flocculation performance in acidic condition, enhanced melting behavior | [62] |
| | Dialdehyde Dextran | Enhanced crosslinking activity which can strengthen the nanostructure of biopolymer-based nanocarriers | |

**Table 4.** *Cont.*

| Polysaccharide | Derivatives | Modification in Properties | References |
|---|---|---|---|
| Gellan | Methacrylated derivatives | Enhanced mucoadhesive property | [89] |
| Starch | Hydroxyalkyl starches | Enhancement in solubility, ease of hydration and swelling power | [23,24] |
| | Acetylated Starch | Imparts hydrophobicity, enhancement of thermoplastic character, retardation of crystallization and lowering of pasting temperature | |
| | Starch cross-linked with sodium trimetaphosphate | Imparts tolerance against extreme pH and high shear conditions | |
| Cellulose | Citric acid cross-linked bacterial cellulose | Improvement in water absorption capacity(About 1.5 times higher) | [90–92] |
| | Carboxymethyl cellulose acetate butyrate | Enhances hydrophobicity and thermoplastic behavior making it suitable for processing as scaffolds | |
| | 1,3 cycloadditions of porphyrin on cellulose | Induction of bactericidal activity | |
| Pullulan | Grafting of methyl acrylate onto pullulan by copolymerization | Increase in hydrophobicity for drug delivery applications | [51,52] |
| | Carboxymethylation of pullulan with sodium chloroacetate | Introduction of negative charge that prolongs the residence time of the polymeric matrix inside the body | |
| Carrageenan | Methacrylated carrageenan | Confers the ability to be photo crosslinked that allows easy tailorability of viscosity, swelling ratio, elastic moduli and pore size distribution | [55] |
| Pectin | Methoxylated derivatives | Altered solubility, gel forming ability, conditions required for gelation, gelling temperature, and gel properties | [93] |
| | Acetylated derivatives | Emulsifying and stabilizing property | |
| | Amidated pectin | Good gelling property and reduced sensitivity against cations and pH | |

## 4. Biomedical Applications of Polysaccharide-Based Implantable Devices

### 4.1. Implants for Oral Cavity

The oral cavity is a significant part of the human body and contributes to the total wellbeing of a person [94]. According to the Global Burden Disease Study conducted in the year 2019, about 3.5 billion people worldwide suffer from oral diseases. The major diseases that affect the oral cavity include periodontal (gum) disease, dental caries (tooth decay), mouth ulcers, and oral cancer [95]. The conventional treatment options available are not sufficient and lack patient compliance, which necessitates the development of some new treatment strategies. Implants have been developed by various researchers for treating diseases of the oral cavity. The polymers used for the preparation of implants for the oral cavity must have mucoadhesive properties so that they can be retained there for a longer duration. They must be biocompatible and biodegradable, and the degradation products of the polymer must be non-toxic and non-immunogenic. Polysaccharides and proteins are the ideal candidates that satisfy these criteria [96].

HA is widely studied for its osteoinductive and osseointegration properties [97]. Surface treatment of HA on titanium (Ti) dental implants enhances migration, proliferation, adhesion, and differentiation of progenitor cells by enhancing the interaction between the bone and the implant (Figure 5a). This facilitates fixation of dental prosthesis precisely in the early loading phase, thus improving patient compliance [98]. Similarly, chitosan has also been reported to have osseointegration capacity. The Ti implants were coated with lactose-modified-chitosan (Chitlac) and this Chitilac-Ti implant was reported to have anti-inflammatory and anti-infective activity [99] (Figure 5b).

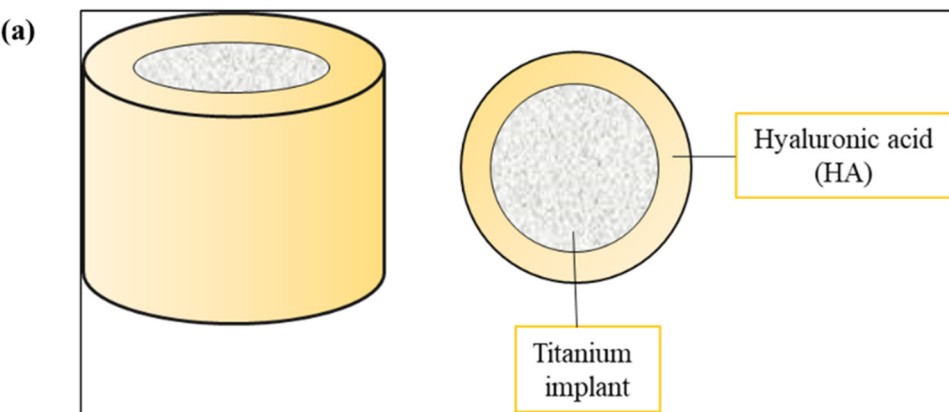

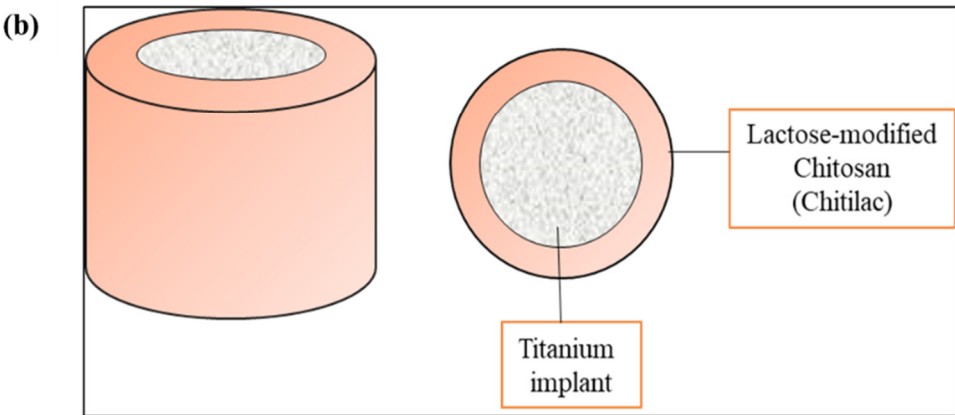

**Figure 5.** Schematic representation of: (**a**) Hyaluronic acid coated titanium implant; (**b**) Titanium implants coated with chitilac (lactose modified chitosan).

Lan et al. synthesised metronidazole loaded composite poly-caprolactone/alginate (PCL/alginate) rings by the solvent casting method, for dental implant (Figure 6a–d). The implant was prepared to deliver metronidazole at a controlled rate for preventing the growth of bacteria at the implantation site. In vitro release studies demonstrated that pure alginate rings showed burst release within a few hours, but the alginate/PCL composite ring showed an initial burst release followed by sustained release for more than 4 weeks after implantation [100].

### 4.2. Implants for Nasal Cavity

Nasal implants are widely used for correction of internal and external nasal valve collapse, in combination also known as lateral wall insufficiency (LWI), leading to nasal obstruction, and also for the treatment of chronic rhinosinusitis (CRS) [101]. The ideal implant for the nasal cavity should be economical, non-toxic, inert, non-carcinogenic, easily available, and should be able to provide mechanical support [102,103]. Various nasal implants have been approved by the FDA, namely, Propel™ implant, Relieva Stratus™

MicroFlow spacer, the Sinu-Foam™ spacer, and many more. "Propel" is a mometasone-releasing PLGA based biodegradable implant approved for the treatment of CRS [104].

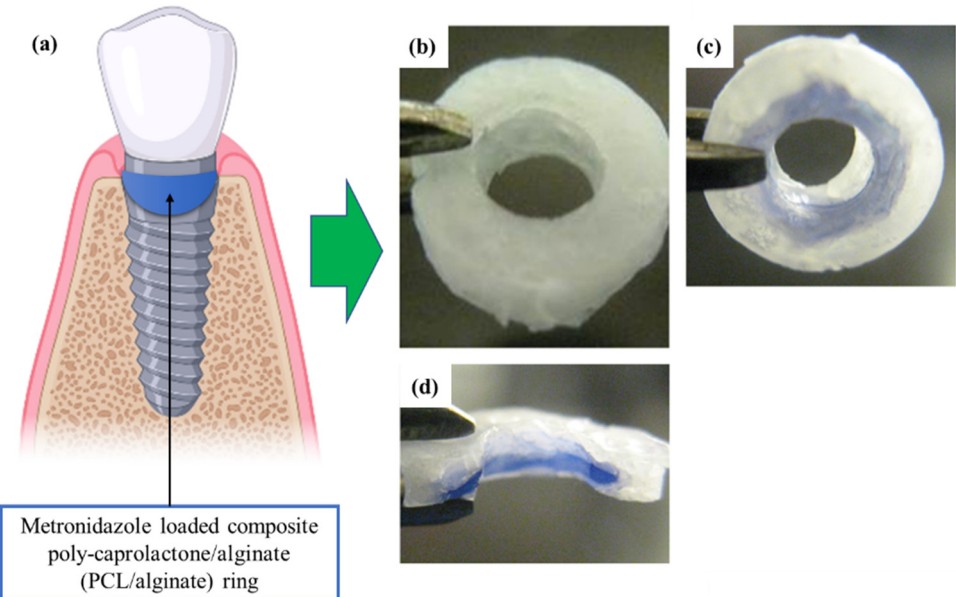

**Figure 6.** Dental implants consisting of (**a**) Metronidazole loaded composite poly-caprolactone/alginate (PCL/alginate) ring; (**b**) PCL ring; (**c**) PCL\alginate ring; (**d**) Central sectional cut of PCL\alginate ring. Reproduced with permission from reference [100].

Silicon (Si) tubes are generally used as an implant for the treatment of the obstruction of nasolacrimal duct, but these implants are found to be associated with side effects such as allergic reactions and bacterial infection, which lead to failure of surgery. In order to overcome these limitations, Park et al. performed hydrophilic polysaccharide based multilayer nanofilm coating on Si-tubes, which has the capability to load as well as release antibacterial and anti-inflammatory agents, i.e., levofloxacin and prednisolone-21-acetate, respectively. They utilized chitosan (CHI) and carboxymethylcellulose (CMC) for the preparation of multilayer films for coating. They observed that CHI/CMC coated Si-tubes exhibited significant antibacterial activity by preventing the attachment of bacteria to them [105]. Figure 7 represents the use of CMC and chitosan coated silicon tube for nasolacrimal duct obstruction.

Feng et al. designed a novel nasal stent comprising of plasticized bacterial cellulose and waterborne polyurethane loaded with the drug poly (hexamethylene biguanide) that exhibited good antibacterial activity. The nasal stent was capable of preventing adhesions and inflammation that can serve as a potential prospect to address postoperative adhesions, infections, and other nasal disorders. Owing to the good water-holding capacity of bacterial cellulose, a moist environment was created on contact with the nasal mucosa and thus wound healing was found to be promoted. The gelation behavior of bacterial cellulose further contributed to the protection of the paranasal sinus mucosa. Thus, the designed nasal stents possess immense clinical applications that can be explored extensively in the near future [106]. Further, in a study, Sang et al. incorporated bacterial cellulose in a 3D-printed gelatin–hyaluronic acid methacryloyl composite hydrogel for the repair of nasal cartilage defects. The bacterial cellulose was found to not only enhance the mechanical properties significantly but also improved the printing fidelity. The biocompatibility of the hybrid constructs was established in ATDC5 cells. Following 7 days, significant proliferation and maintenance of the expression of specific proteins was evident making it a promising candidate for nasal cartilage repair [107]. Bleier et al. studied the effect of antibiotic impregnated chitosan glycerophosphate implant in the treatment of acute bacterial sinusitis. The studies concluded that in a rabbit model, a greater log reduction of colony forming units (CFU) of gram-positive and gram-negative bacteria was attained through the

antibiotic impregnated implant in comparison to daily antibiotic irrigations [108]. Further, in another study, dexamethasone loaded biodegradable stent composed of chitosan glycerophosphate was developed for intranasal implantation. A measure of 91.7% deacetylated chitosan glycerophosphate loaded with the drug was implanted into the maxillary sinuses of 12 rabbits. The inert, malleable, semi-rigid chitosan glycerophosphate insert was capable of eluting the steroid for over 15 days showing negligible signs of inflammation. Thus, lack of inflammatory potential of the polymer makes it a promising stent loaded with pharmacologically active agent for future biomedical applications [109].

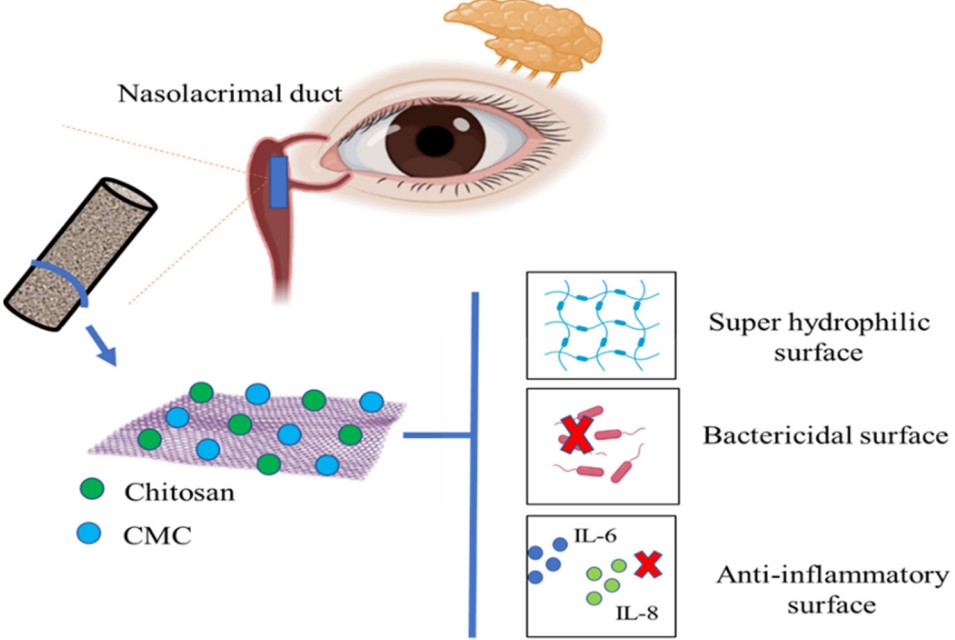

**Figure 7.** Schematic representation of use of CMC and chitosan coated silicon tube for nasolacrimal duct obstruction. Adapted with permission from reference [105]. Copyright 2022, Elsevier.

### 4.3. Bone Implants

Osteointegration between the implanted biomaterial and the surrounding bone is critical for the acceptance of implants by the human body as it eliminates the outgrowth of fibrous tissue at the bone-implant interface [110]. Polysaccharide-based biomaterials offer good potential in the treatment of critical-sized bone defects due to their tailorable chemical and biological properties. Chitosan based scaffolds are widely researched for tissue engineering purposes. Lyophilization of chitosan acetate solution results in the formation of porous interconnected structures that are ideal for cell seeding, cell migration, and nutrient supply that facilitate bone regeneration. Electrospinning, particle aggregation, and solvent-exchange phase separation are other methods employed in the generation of chitosan scaffolds (Figure 8) [111].

Michael et al. synthesized electrospun hydroxyapatite-containing chitosan scaffolds for bone tissue engineering to overcome the drawbacks associated with autografts in the treatment of large bone defects. Autografts exhibit poor integration with the host's bone due to the absence of periosteum, the outer layer of bone that recruits mesenchymal progenitor cells for bone remodelling. The chitosan-based scaffolds were found to act as a bridge at the interface of the periosteum, promoting the adhesion, proliferation, and differentiation of 7F2 osteoblast-like cells [112]. Figure 9 demonstrates the use of chitosan based implantable scaffold for osteointegration.

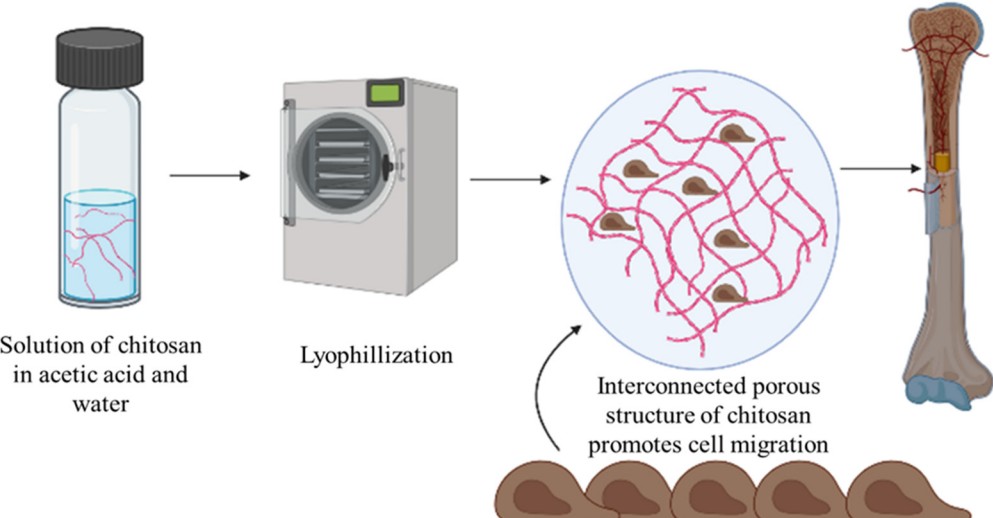

**Figure 8.** Diagrammatic representation of chitosan based implantable scaffold for bone regeneration.

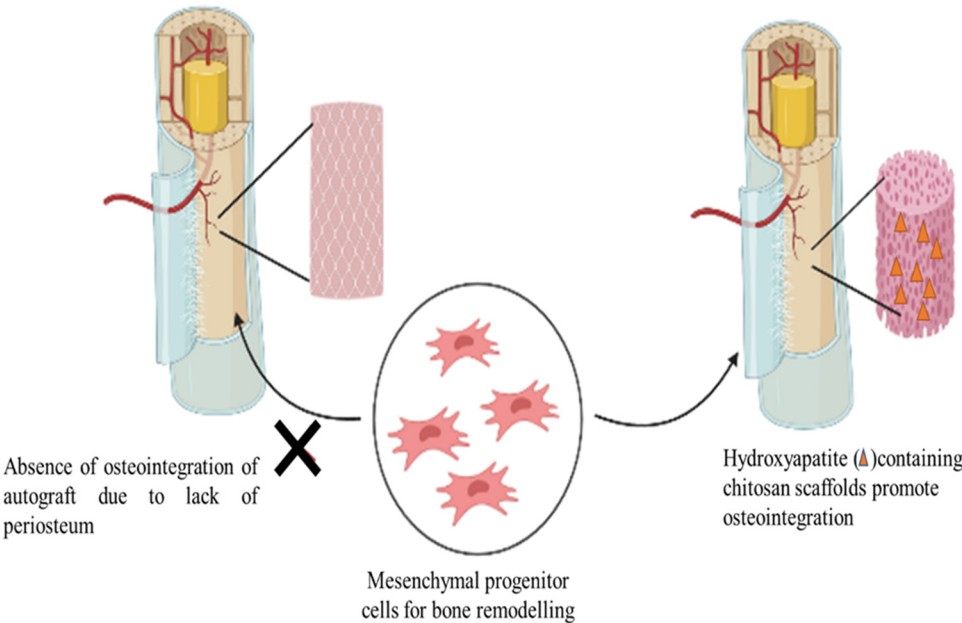

**Figure 9.** Implantable scaffold for effective osteointegration in bone.

Promotion of bone regeneration can be achieved by the delivery of therapeutic agents such as growth factors via implantable biomaterials. Lee et al. developed a chitosan–silica hybrid membrane for the delivery of bone morphogenetic protein-2 (BMP-2) and evaluated the bone healing capacity using in vivo and in vitro studies (Figure 10). BMP-2 exhibited excellent affinity towards the hybrid membrane due to its mesoporous structure. The efficacy of the membrane to act as a carrier was established by evaluating the induction of BMP-2 mediated cellular responses such as proliferation and differentiation in cell-culture studies. In vivo studies also indicated that a short-term implantation of the hybrid membrane for about 2 weeks accelerated the healing of bone defects [113].

Tissue engineering of non-load bearing bones, such as the trabecular, maxillofacial, or craniofacial bones, involves the use of bio-polymeric scaffolds owing to their definite microarchitecture and the ability to alter the spatio-temporal distribution of therapeutic molecules at the injury site. Agarwal et al. designed a novel alginate bead-based 3D implant using metronidazole as the model drug against bone infections caused by *E. coli*. Hexagonal close packed layers of calcium alginate beads were stacked to produce a patterned array of interconnecting octahedral and tetrahedral pores. The respective average diameter of the

pores was found to be 262.9 and 142.9 μm. A 2.7-fold increase in the compressive modulus was observed on incubation of the implant in simulated body fluid. The increase in the rigidity of the implant with time could be attributed to the progressive ionotropic gelation of the alginate molecules. The osteoconductive nature of the implant was confirmed through in vitro studies, in which increased expression of differentiation markers such as runx2, alkaline phosphatase, and collage type 1 was observed in human mesenchymal stem cells [114].

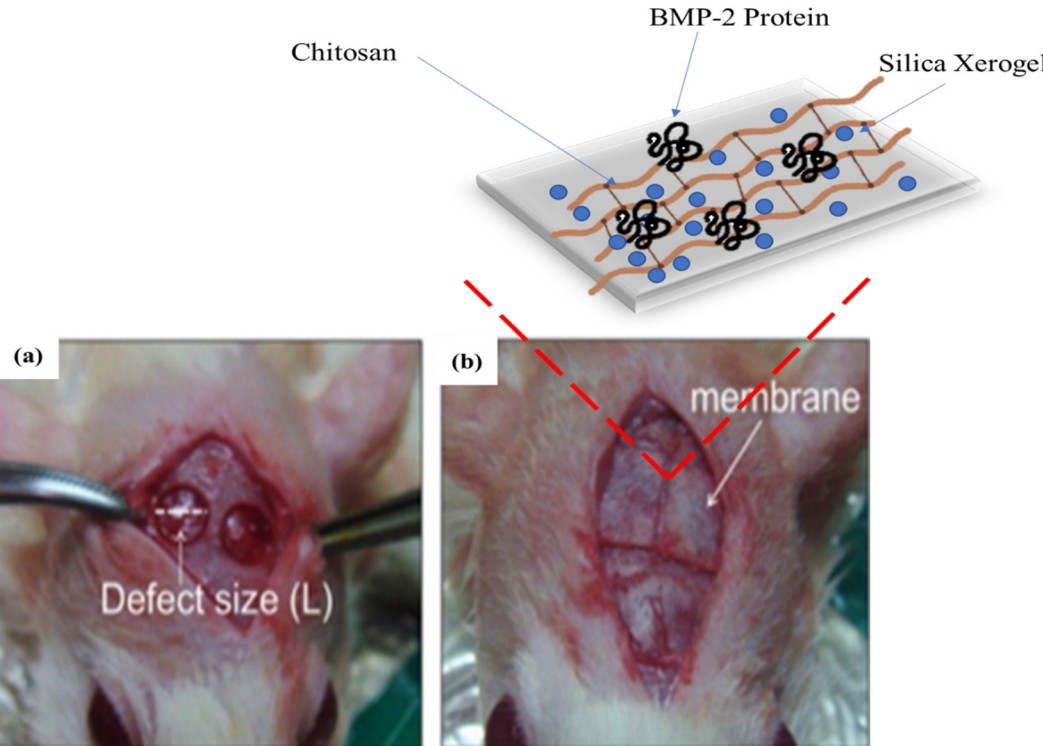

**Figure 10.** (**a**) Calvarial bone defect model; (**b**) Chitosan–silica hybrid membrane for the delivery of bone morphogenetic protein-2. Adapted with permission from reference [113]. Copyright 2022, Elsevier.

In order to mimic the bone's natural extracellular matrix, Minhao et al. constructed an epichlorohydrin-crosslinked hydroxyethyl cellulose (HEC)/soy protein isolate (SPI) scaffold, for the repair of large-bone defects. HEC promotes the attachment and growth of human mesenchymal stem cells, making it ideal for bone tissue engineering applications. Soy protein isolate, owing to the presence of polar functional groups such as carboxyl, hydroxyl, and amine groups, is easily tailorable to obtain various mechanical properties. Hydroxyapatite functionalization conferred targeting properties to the scaffold. The calcium/phosphorus ratio of the scaffold (1.65) containing 70% SPI, was similar to that of natural bone tissue (1.67). The expression of osteoporosis related genes was found to be upregulated during MC3T3-E1 cell differentiation studies [115]. Modified cellulose derivatives such as cellulose acetate films and membranes, apart from being widely used as scaffolds, are nowadays being explored as potential protective coatings and drug delivery vehicles for implants. Faria et al. successfully incorporated microcapsules of the drug daptomycin into cellulose acetate films, which were coated on to stainless steel implants owing to the good adhesion properties and biocompatibility of cellulose fibers. Adhesion of cellulose films to the implant was further enhanced by electro spraying an intermediate layer of chitosan. The concentration of drug released after 120 min was found to be $3 \times 10^{-3}$ mg/mL. Hence, multifunctional polysaccharide-based films can effectively reduce the formation of biofilms at the interface of implants due to their ability to release the drug in a controlled fashion [116].

The extrusion-based 3D printing technology is widely used in the fabrication of artificial bone graft substitutes that overcome the donor site complications inherently associated with autologous grafts. Bhattacharjee et al. developed a $Zn^{2+}$ functional-ized hydroxyapatite-starch composite for orthopedic applications. The poor mechanical strength of starch was hypothesized to be overcome by the formation of a zinc–starch complex. Experimental results revealed that a four-fold increase in compressive strength was achieved upon $Zn^{2+}$ functionalization. The functionalized grafts maintained mechanical integrity throughout the 6-week dissolution study in simulated body fluid, whereas the non-functionalized HA-starch grafts were found to degrade within a week [117].

Owing to the limited repair capacity of articular cartilage, tissue engineering is a promising approach to regenerate osteochondral defects. In this, cells are extracted from the patient and cultured in vitro to achieve a sufficient number of cells and extracellular matrix. The resultant mass is then incorporated in a three-dimensional scaffold and further implanted at the site of defect. However, the lack of in vivo integration of the engineered cartilage and the subchondral bone is a major limitation. In order to overcome this, Ghosh et al. developed bi-layered scaffolds by compression molding and subsequent particulate leaching to enhance in vivo integration of the engineered cartilage tissue. The cartilage side of the scaffold was composed of starch and poly (L-lactic acid) (PLLA), whereas hydroxyapatite and PLLA constituted the side exposed to the bone, providing enhanced stiffness and strength. The presence of starch in the cartilage side confers it the ability to uptake adequate amount of water to mimic the water content (around 80%) of the human articular cartilage. The pore size in the bone region of the construct was found to be higher than the cartilage to promote vascularization. Scanning electron microscope images revealed the presence of interconnecting structures across the two layers that are essential for the diffusion of nutrients and waste products. Hence, bilayered scaffolds exhibit promising potential in tissue engineering applications for successful host integration [118].

Fricain et al. developed composite scaffolds of hydrophilic polysaccharides pullulan and dextran, supplemented with hydroxyapatite particles that are nanocrystalline in nature. In vitro studies on human bone marrow stem cells, in a medium lacking osteoinductive factors, were carried out, and the scaffolds were found to induce the expression of early and late bone specific markers. On heterotopic subcutaneous and intramuscular implantation in mice and goats, respectively, the scaffold was able to induce the deposition of a layer of hydroxyapatite and retain the local growth factors such as VEGF165 and BMP-2. Formation of a dense mineralized tissue in the case of mice and an osteoid tissue in the case of goats was observed. Further, enhancement of bone tissue regeneration was observed on orthotopic implantation of the scaffold in three different sites, namely, the femoral condyle, tibial osteotomy, and transversal mandible, thus demonstrating the wide applicability of the matrix in orthopedic and maxillofacial surgical applications [119].

The polyelectrolyte multilayer scaffolds of chitosan/iota-carrageenan and chitosan/pectin have been investigated for their potential use as a cyto-compatible, antibacterial coating in implants. In vitro tests confirmed the bactericidal activities of the scaffolds against *Staphylococcus aureus* (gram-positive) and *Pseudomonas aeruginosa* (gram negative). The low water contact angle (25°) of the chitosan/iota-carrageenan assembly confers a suitable microenvironment for the adhesion and proliferation of bone marrow stem cells. Thus, the polyelectrolyte multilayer scaffolds act as a suitable biomaterial to promote healing around orthopedic implants and can be widely used in applications that necessitate protection against bacterial proliferation [120]. The emerging innovations in bone-tissue engineering are thus largely driven by the new generation of modified natural biomaterials. The comprehensive scope of research in the field of polysaccharide-based bone implants could lead to its wide clinical use in the near future.

*4.4. Implant for Ocular Use*

Proteins and polysaccharides appear as ideal candidates for biodegradable drug delivery due to their biocompatibility, biodegradability, low immunogenicity, and pH

stability under physiological conditions. Polysaccharides such as cellulose, hyaluronic acid, gelatin, collagen, xanthan gum, alginic acid, and chitosan have been successfully explored in drug delivery for eye diseases. These polysaccharides are extensively used as additives for improvements in permeability, contact time, and ocular absorption. Chitosan is the most widely explored polymer for ocular drug delivery due to its mucoadhesive property and inertness [121]. In one study, Manna et al. prepared intravitreal chitosan and polylactic acid-based methotrexate micro-implants to treat primary intraocular lymphoma. The results from their study indicate that uncoated chitosan methotrexate implants administer drug approximately for 1 day, and after coating with polylactic acid, the implants show drug release for 50 days with a release rate of 0.2–2.0 µg/day [122]. In further continuation of their previous work, to improve the methotrexate release profile, Manna et al. utilized different combinations of PLGA-PLA coating. They observed two findings after the increase in the PLA content in PLGA: (a) the initial burst release effect gets reduced, and (b) delayed swelling and biodegradation of the micro implants. After coating with different ratios of PLA-PLGA, they observed drug release of 0.2–2.0 µg/day of methotrexate for an extended period of ∼3–5 months [123].

There is a cellulose-based FDA approved implant of fluocinolone acetonide for the treatment of chronic non-infectious uveitis also available in the market named Retisert®. This implant is a fluocinolone acetonide tablet containing microcrystalline cellulose, magnesium stearate, and PVA with a coating of silicone elastomer coating with a small orifice for drug release. A PVA based semi-permeable layer is placed between the tablet and the release orifice to create a diffusion barrier for drug release. Clinical trial study results showed that implants reduced the chances of disease recurrence, the need for supplementary topical, systemic, and periocular therapies, and increased visual acuity [124,125]. There is another FDA-approved insert of hydroxypropyl cellulose available in market for the treatment of patients with moderate to severe dry eye syndromes, including keratoconjunctivitis sicca. LACRISERT® (hydroxypropyl cellulose ophthalmic insert) is a rod-shaped, translucent, sterile, water soluble, ophthalmic insert, for administration into the inferior cul-de-sac of the eye (Figure 11). LACRISERT® works by stabilizing and thickening of precorneal tear film and prolong the breakup time of tear film, which is usually accelerated in patients with dry eye states. Along with this, LACRISERT® also protects the eye by providing lubrication [126].

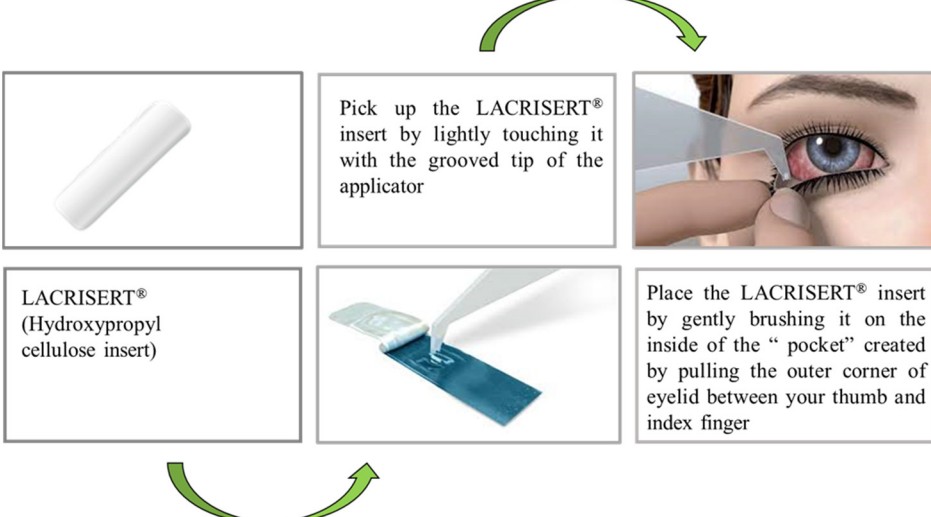

**Figure 11.** Protocol for administration of LACRISERT® into eye (https://www.lacrisert.com/) (accessed on 28 July 2022).

*4.5. Implants for Antiviral Therapy*

Viruses, as evident from the past, are known to cause diseases that have the potential to cause an epidemic/pandemic. Major viral diseases that are prevalent around the world include Human immunodeficiency virus infection and acquired immune deficiency syndrome (HIV-AIDS), hepatitis, Ebola, and respiratory pathogens like the human influenza severe acute respiratory syndrome and middle east respiratory syndrome [127]. According to World Health Organization reports, an estimated 37.7 million people around the world were found to be infected with HIV by the end of the year 2020. Up till now, no cure has been available for HIV infections. However, the disease can be managed by timely diagnosis, treatment, and care, enabling infected people to live a long and healthy life [128]. The major limitation associated with antiretroviral therapy is its longer duration of treatment, leading to nonadherence to the medication [129]. In order to overcome these limitations, long-acting antiviral drug-loaded biodegradable implants have been developed which can offer sustained release of the antiretroviral drug for a considerable period of time, ranging from several weeks to months [130]. Though, the utilization of polysaccharide based coated implants for antiretroviral therapy is not yet established in the literature, various polysaccharides including heparin, galactan, fucoidan, glucan, cellulose, dextran, or dextrin have been reported to possess antiviral properties, that can be explored in the future for prevention of viral infections [131]. The layer-by-layer coating of polysaccharides on the implant surface can be used as a potential approach to prevent viral growth on implants [132]. The ability of sulfated polysaccharides to mimic the glycosaminoglycans present in the cell membrane confers it with distinct antiviral properties. Sulfated polysaccharides are known to interfere with the steps involved in the lifecycle of a virus such as adsorption, invasion, transcription, and replication and thus lead to an enhanced host immune response by accelerating the viral clearance rate. Hence, they offer a potential for further scientific and clinical research on implantable systems [133].

## 5. Regulatory Considerations for Implantable Device

Regulatory agencies are consumer protection agencies that promote and protect human health. Biomaterials such as polysaccharides used in the field of drug delivery and regenerative medicine are generally regulated as medical devices. Medical devices pose numerous challenges for the regulators as the safety and effectiveness of these biomaterials should be unquestionable which in turn necessitates creation of proper regulatory frameworks.

In the United States, the design, large-scale production and commercialization of medical devices is regulated by the Center for Devices and Radiological Health (CDRH), a branch of the Food and Drug Administration. Medical devices are classified as Class I, II, and III based on the significant risks posed by the device. Class I devices pose the lowest risk and include tongue depressors, external dressings, etc., that require only general controls to establish its safety and efficacy. However, implantable drug delivery systems, owing to their direct and prolonged interaction with the host tissue usually fall under Class II and III that correspond to an intermediate and a significantly higher risk level, respectively [134]. Thus, it requires special control that includes biocompatibility data, device characterization as per FDA recognized standard, product-specific controls, special labelling, and performance standards. Implantable drug delivery systems require the FDA's approval of a pre-market application [510 (k)] in order to be marketed in the United States. In order to gain approval, the developer needs to demonstrate substantial equivalence to an already marketed device in the United States by in vitro and animal studies. If no equivalence to an existing product can be established, then the new device requires demonstration of safety and efficacy through clinical studies to gain approval of 510 (k) pre-market application [135]. Clinical studies can be further conducted if only an Investigational Device Exemption (IDE) is submitted and thereby approved by the FDA. An IDE must consist a detailed description of the product, methods of manufacturing, and surgical protocols. The data obtained from pre-clinical studies, conducted as per Good Laboratory Practices (GLP), have to be furnished and characterization of the product

must be in accordance to the guidelines and recognized standards. An IDE is considered approved after 30 days of submission to the FDA unless the agency informs the sponsor otherwise. The agency disapproves the application if the associated risks of the device outweigh its benefits. Approval of the pre-market application is based on risk-benefit assessment and if approved, the sponsors can begin the clinical trial.

In the case of combination products such as drug–device, device–biologic, and drug–biologic–device that comprise of two or more components, the process of jurisdiction is critical. In the United States, biomaterials are regulated by CDRH, biologics are regulated by the Center for Biologics Evaluation and Research (CBER) and drugs are evaluated by the Center for Drugs Evaluation and Research (CDER). The most crucial therapeutic effect of the product is identified which decides the appropriate division that will lead the regulatory pathway in the review process [136].

The FDA has established Quality System Regulations (QSR) 21 CFR 820.30 to guide medical device manufacturers in the design, production, packaging, and distribution of the product. Periodic inspections are conducted by the FDA to assure that the manufactures meet the expected cGMP standards [137].

### 5.1. Important Laboratory Testing Required for Approval

#### 5.1.1. Material Characterization

The parameters that are crucial for characterization of tissue engineering biomaterials include mechanical, morphological, and chemical properties. Mechanical performance that is required is based on the site of implantation and the corresponding physiological environment. The scaffold must provide sufficient mechanical support to allow proper regeneration of the tissue. For example, high compressive strength is required for a bone scaffold whereas a vascular scaffold requires high burst strength [136]. Further, the chemistry of the biomaterial has to be extensively characterized for identification of any potential toxic, carcinogenic, and immunogenic effects. The molecular weight of polymers also needs to be characterized as it decides the processing steps of the biomaterial. Physical properties such as pore size, porosity, architecture, and interconnectivity need to be characterized [138].

#### 5.1.2. Biocompatibility

The FDA guidance document on the use of International Standard ISO 10993-1 "Biological evaluation of medical devices—Part 1: Evaluation and testing within a risk management process" helps in the determination of any possibilities of an unacceptable adverse biological response as a consequence of contact of the biomaterial with the human body. The biocompatibility test to be done depends on the intended use and category of the device. Cytotoxicity tests are generally performed to assess cell death or any other potential adverse effects on the cells. Other general tests include acute systemic toxicity, sensitization, hemocompatibility, pyrogenicity, genotoxicity, carcinogenicity, and reproductive toxicity. The implantation test is performed to estimate the local effects caused by the test material post implantation, by comparing the tissue response to the control [139].

#### 5.1.3. Sterility

Sterilization ensures patient safety upon implantation owing to the nonexistence of viable microorganisms on the product. Terminal sterilization through gamma irradiation or ethylene oxide sterilization is recommended by the FDA. Packaging system is a crucial parameter to achieve effective terminal sterilization as it must allow permeation of the gas/radiation to reach the biomaterial. Equipment qualification, bioburden testing, and sterility testing are the key requirements as per the FDA [139].

Further, data obtained from pre-clinical studies need to be provided. Following clinical investigation, the approval of the pre-market application is completely based on assessment of risk-benefit ratio obtained from the trials. The aforementioned regulatory requirements have posed many challenges for products seeking market approval. Hence, extensive

testing with proper quality management is recommended for successful marketing of these biomaterials.

## 6. Conclusions and Future Perspective

The use of polysaccharide-based implantable devices in the treatment of various diseases is becoming increasingly important. Polysaccharides are used in the development of implantable devices to improve its biodegradability and biocompatibility. In addition, polysaccharides also confer certain unique properties to the composites such as mechanical strength, which favors the tissue reconstruction process. This review discusses the examples of several polysaccharides, with a special focus on the physical features of these polymers that enables them to perform the designated function in the body. Furthermore, their potential utility in multiple implantable devices in various diseased states has been demonstrated. Though only a few commercial products, as discussed in the previous sections, have been successfully developed, the scope of this field is emerging vastly and holds a promising potential to create a niche market. In recent times, a considerable number of efforts have been devoted towards the development of biodegradable polysaccharide implants by various researchers and it can be expected in the near future that these innovative composites can undergo scale-up and commercialization. This would serve as a breakthrough achievement in the field of biomedical sciences, thus expanding the scope of tissue engineering applications. Polysaccharide-based implanted devices or coated implants outperform synthetic or semi-synthetic polymers. Polysaccharide-based devices are now being studied/explored for their physicochemical features, which include surface morphology, in-vitro characterization, and in-vitro evaluation. However, once implanted as a medical intervention, the implants begin to integrate the unique interaction with human body elements such as cells, tissue, organs, or the endocrine system. As a result, it is critical to comprehend such a potential interaction and research the side effects of those implantable devices. Because polysaccharides are widely used in biomedical and pharmaceutical applications, further examination is required due to safety concerns. Polysaccharide-based materials are now regarded safe in terms of biocompatibility, biodegradability, and non-toxicity, although additional research should be conducted. Once, the safety of these devices is well-established, it will in turn enhance the patient acceptability. Further, the degradation rate and the mechanism of degradation has to be well-studied for each polysaccharide. The erosion rate can significantly affect the drug release. Quick degradation in the physiological environment and result in excessive release of the drug (burst release) and may also result in premature loss of strength of the polysaccharide. Thus, the degradation mechanism of each polysaccharide needs to be validated and must be well controlled so that a better idea can be obtained regarding its in-vivo behavior. Further, a thorough understanding of the impact of the physiological microenvironment on the degradation behavior of each polymer is required for the design of successful products. For example, an acidic microenvironment may catalyze the degradation owing to acid-mediated hydrolysis of the functional groups that constitute the backbone of the polysaccharide. Owing to the complexity of the human physiology, the translational potential of these materials needs extensive research in the long-term process. Every country has its own regulatory authority, and each country has created specific requirements for using such materials in biomedical applications and pharmaceutical use. However, it is crucial to develop worldwide legislation for the manufacture, utilization, and safety of biomaterials such as polysaccharides. This would aid in the development of more regulatory standards, utilization, and clinical trials in polysaccharide-based implantable devices in the future.

**Author Contributions:** S.S.: conceptualization, writing, and editing; D.R.: writing and editing; A.S.: writing and editing; K.B.: writing and editing; R.G.: writing and editing; S.K.: writing and editing; D.B.: review, editing, visualization, and supervision; N.K.: review, editing, visualization, and supervision. All authors have read and agreed to the published version of the manuscript.

**Funding:** This research received no external funding.

**Institutional Review Board Statement:** Not applicable.

**Data Availability Statement:** Not applicable.

**Acknowledgments:** We would like to sincerely thank NIPER-A for providing us with all resources.

**Conflicts of Interest:** The authors declare no conflict of interest.

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
