# Peer review of "Polysaccharide Based Implantable Drug Delivery: Development Strategies, Regulatory Requirements, and Future Perspectives"

_2673-4176, doi:10.3390/polysaccharides3030037_

Round 1
Reviewer 1 Report
This is a nice review in which the authors summarized the developments of polysaccharide-based implantable drug delivery. In particular, they introduce several polysaccharide polymers such as starch, cellulose, alginate, chitosan, pullulan, carrageenan, dextran, hyaluronic acid, agar, pectin, gellan gum. I would like to recommend the publication of the paper after minor revision.
Some comments:
The authors are suggested to discuss the pros and cons of starch, cellulose, alginate, chitosan, pullulan, carrageenan, dextran, hyaluronic acid, agar, pectin, gellan gum in terms of their implantable drug delivery applications.
The authors need to design a table to summarize the content of the review.
The authors need to propose more future research directions in this field.
The authors should carefully search the literature, and add some missing references in this field.
Author Response
We appreciate the reviewer for the valuable suggestion to improve the quality of manuscript. As per the suggestions we modified the manuscript.
Comment 1: The authors are suggested to discuss the pros and cons of starch, cellulose, alginate, chitosan, pullulan, carrageenan, dextran, hyaluronic acid, agar, pectin, gellan gum in terms of their implantable drug delivery applications.
Response: As suggested, the pros and cons of starch, cellulose, alginate, chitosan, pullulan, carrageenan, dextran, hyaluronic acid, agar, pectin, gellan gum in terms of their implantable drug delivery applications have been summarized in table 3 of the revised manuscript.
Comment 2: The authors need to design a table to summarize the content of the review.
Response: As per the reviewer’s suggestion, it has been done in the revised manuscript (Table 3).
Comment 3: The authors need to propose more future research directions in this field.
Response: More future directions have been included in the revised manuscript and highlighted in yellow color (Section 6).
Comment 4: The authors should carefully search the literature and add some missing references in this field.
Ans: As per the reviewer’s comment, some missing references have been included in the revised manuscript.
Reviewer 2 Report
The work reviewed the application of polysaccharides in the development of implantable drug delivery. After some minor additions, it could be considered for publication.
1. Add the tables for easier overview.
2. The commercial products should be added more for an actual landscape of this field.
Author Response
We appreciate the reviewer for valuable suggestion to improve the quality of the manuscript.
Comment 1: Add the tables for an easier overview.
Response: As per suggestion Tables 1, 3, and 4 have been newly added to the revised manuscript.
Comment 2: Commercial products should be added more for an actual landscape of this field.
Response: Available commercial implants have been added to the revised manuscript (table 1) and highlighted in yellow color.
Reviewer 3 Report
In this work, Salave et al. reviewed the implantable drug delivery based on polysaccharide materials, introducing the development strategies, regulatory requirements, and future perspectives. It may eventually be published but requires major revisions as indicated.
1. The content of this review is inconsistent with the title. While the major part of this review is to introduce the materials and biomedical applications of polysaccharide-based implantable drug delivery systems, only a few focus on development strategies, regulatory requirements, and future perspectives.
2. It would be better to provide the tables in Part 3, which compare the advantages, disadvantages and properties of different polysaccharide-based polymers in various applications.
3. The key point of this review is to review the polysaccharide materials for implantable drug delivery applications and provide potential readers with an understanding of the current status of biomedical translation. Part 2 and 4 should be added to improve the authors’ opinion of the work.
4. Some grammatical mistakes and slips in the texts. Please check the manuscript carefully.
Author Response
We appreciate the reviewer for the valuable suggestions to improve the quality of review. As per the suggestion we modified the manuscript.
Comment 1: The content of this review is inconsistent with the title. While the major part of this review is to introduce the materials and biomedical applications of polysaccharide-based implantable drug delivery systems, only a few focus on development strategies, regulatory requirements, and future perspectives.
Response: As per the reviewer’s suggestion, development strategies, regulatory requirements, and future perspectives have been discussed in detail in the revised manuscript and highlighted in yellow color (Section 5 and 6).
Comment 2: It would be better to provide the tables in Part 3, which compare the advantages, disadvantages, and properties of different polysaccharide-based polymers in various applications.
Response: As per suggestion, table 3 has been included for the comparison of advantages and disadvantages of polysaccharide-based polymers whereas table 4 has been included for the properties of different polysaccharide-based polymers.
Comment 3: The key point of this review is to review the polysaccharide materials for implantable drug delivery applications and provide potential readers with an understanding of the current status of biomedical translation. Parts 2 and 4 should be added to improve the authors’ opinion of the work.
Response: As per the reviewer’s suggestion, part 2 and 4 has been improved. In part 2 detailed strategies for implant development have been included in the revised manuscript and highlighted in yellow color. Further, in part 4 missing reference has been included.
Comment 4: Some grammatical mistakes and slips in the texts. Please check the manuscript carefully.
Response: As per the reviewer’s comments, the whole manuscript has been checked for language corrections.
Round 2
Reviewer 1 Report
The authors have addressed my comments.
Reviewer 2 Report
The work was improved and considerable for publication.
Reviewer 3 Report
Accept at present state